# Superior Photocatalytic Activity of BaO@Ag₃PO₄ Nanocomposite for Dual Function Degradation of Methylene Blue and Hydrogen Production under Visible Light Irradiation

**Hanaa Selim** [1], **E. R. Sheha** [2], **Rania Elshypany** [1], **Patrice Raynaud** [3], **Heba H. El-Maghrabi** [3,4],* and **Amr A. Nada** [1,3],*

[1] Department of Analysis and Evaluation, Egyptian Petroleum Research Institute, Cairo P.O. Box 11727, Egypt
[2] Cyclotron Facility, Nuclear Physics Department, Nuclear Research Center, Egyptian Atomic Energy Authority, Cairo P.O. Box 13759, Egypt
[3] Laboratoire Plasma et Conversion d'Energie (LAPLACE), Université de Toulouse, CNRS, INPT, UPS, 31062 Toulouse, France
[4] Department of Refining, Egyptian Petroleum Research Institute, Cairo P.O. Box 11727, Egypt
* Correspondence: hebachem@yahoo.com (H.H.E.-M.); chem_amr@yahoo.com or amr.nada@toulouse-inp.fr (A.A.N.)

**Abstract:** The current work focuses on the photo degradation of organic pollutants, particularly methylene blue (MB) dye, and the production of hydrogen as green energy using a composite of silver phosphate $Ag_3PO_4$ (AP) and barium oxide/silver phosphate $BaO@Ag_3PO_4$ (APB) as a photocatalyst. This composite was successfully synthesized using a chemical co-precipitation approach. The physicochemical properties of the obtained samples were investigated using scanning electron microscopy (SEM), energy dispersive X-ray spectroscopy (EDX), X-ray diffraction (XRD), Fourier-transform infrared spectroscopy (FT-IR), ultraviolet–visible diffuse reflectance spectroscopy (UV–Vis/DRS), and photoluminescence (PL) spectrophotometry. From XRD, the average crystallite sizes of AP and APB are 39.1 and 46 nm, respectively, with a homogeneous morphology detected by SEM. UV and PL experiments showed that the compound is active under visible light, with an improvement in the lifetimes of the electrons and the holes in the presence of BaO with $Ag_3PO_4$. The as-synthesized APB photocatalyst sample showed a remarkably high degradation efficiency of MB (20 ppm, 50 mL) of around 94%, with a hydrogen production yield of around 7538 μmol/(h·g), after 120 min of illumination, which is greater than the degradation efficiency of the AP photocatalyst sample, which was about 88%. The high photodegradation efficiency was attributed to the electronic promotion effect of the BaO particles. The APB composite demonstrated an increased photocatalytic performance in effectively degrading an organic dye (MB) with no secondary pollutants when exposed to visible light irradiation.

**Keywords:** hydrogen production; water treatment; silver phosphate ($Ag_3PO_4$); barium oxide (BaO); organic pollutant; photocatalytic degradation

## 1. Introduction

Water treatment is becoming a major global concern due to a scarcity of clean drinking water, a considerable growth in the population [1], and an increase in water-polluting industries such as oil refineries, dye factories, and hospitals [2]. There are three major types of waste pollutants: biological contaminants (bacteria, viruses, and fungi), mineral pollutants (sulphate, nitrate, and heavy metals), and organic contaminants (dung, medicinal, dyestuffs, oil, and parasites) [3]. Effective methods for removing organic compounds from water have sparked considerable interest. For the removal of organic pollutants from polluted water and wastewater, a variety of techniques, including coagulation, filtration with coagulation, precipitation, ozonation, adsorption, ion exchange, reverse osmosis, and advanced oxidation processes (AOPs), have been used.

The photocatalysis and AOPs subset is an environmentally benign method that utilizes clean sunlight to degrade organic pollutants in water. In another are of attention, hydrogen production is one of the most promising renewable energy sources. Its nature is eco-friendly (carbon-free fuel), limitless, efficient, and cost-attractive [4]. Meanwhile, environmental problems resulting from the combustion of non-renewable fossil fuels have compelled scientists to focus on exploring green and renewable alternative energy sources. Solar-driven hydrogen production has drawn considerable interest, as solar energy is a clean, inexhaustible energy resource. Due to the daily and seasonal variability of sunlight, the energy harvested from the sun needs to be efficiently converted into clean chemical fuel that can be stored, transported, and used upon demand.

Photocatalytic hydrogen generation is a clean, eco-friendly, and renewable source. Researchers are interested in semiconductor-based photocatalysts because they exhibit a high efficiency when exposed to visible light. Studies have shown that, when compared to the majority of other known photocatalysts, these photocatalysts have significantly improved photo-oxidative capabilities and a higher photocatalytic degradation efficiency [5]. Among of them, silver orthophosphate ($Ag_3PO_4$) is currently an important host material for activator ions in its lattice because of its exceptional chemical stability, greater product yield, and low annealing temperature, and because it has a suitable band gap of approximately 2.45 eV [6]. Furthermore, it has been identified as a promising photocatalyst activated by visible light irradiation and shows strong photo-oxidative activity for the production of $O_2$ from water through the splitting and degradation of the organic pollutants as dyes [7]. However, there is one major disadvantage to applying pure $Ag_3PO_4$, and that is its sensitivity to photo illumination, as a result of the photo-generated corrosion electrons [8]. To overcome this limitation, $Ag_3PO_4$ has been conjugated with other semiconductor materials such as $SnO_2$ [9], AgBr [10], $In(OH)_3$ [11], $CeO_2$ [12], $TiO_2$ [13], $NiFe_2O_4$ [14], and $ZnFe_2O_4$ [15]. Another material that has received extensive interest as a means to improve the photo stability and photoactivity of $Ag_3PO_4$ is barium oxide (BaO) [16]. Whereas (BaO) is frequently used to improve catalyst activity and has previously been used as a dopant for producing efficient and reusable photocatalysts, it is also widely used in many applications such as artefacts, wall paintings [17], the biomedical field [18], catalysts [19], the pharmaceutical industry [20], radiation dosimetry [21], etc.

The work presented herein focused on the visible-light-driven photo degradation of a dye pollutant, in particular methylene blue (MB) dye, and on hydrogen production using the nanostructure systems of $Ag_3PO_4$ and $BaO@Ag_3PO_4$ (APB) compounds as a photocatalyst. Furthermore, the physicochemical properties of the obtained samples were extensively studied to explore their excellent photocatalytic activity for the degradation of MB under visible light irradiation and their greater rate of hydrogen production as compared to pure $Ag_3PO_4$. The synthesised APB proved to be an effective photocatalyst material for applications related to environmental protection.

## 2. Results and Discussion

The physicochemical properties of the obtained samples were characterized using scanning electron microscopy (SEM), energy dispersive X-ray spectroscopy (EDX), X-ray diffraction (XRD), Fourier-transform infrared spectroscopy (FT-IR), ultraviolet–visible diffuse reflectance spectroscopy (UV–Vis/DRS), and photoluminescence (PL) spectrophotometry.

### 2.1. Structural Analysis

#### 2.1.1. Scanning Electron Microscopy (SEM) Analysis

The surface morphologies of the prepared materials were analyzed using SEM. In order to observe the morphology of the composite sample, SEM images of the AP and APB samples were recorded and are shown in Figure 1. The SEM image of pure $Ag_3PO_4$ presents a nanorod-like structure with a small spherical particle on the surface (Figure 1a). In the presence of BaO, these plates became smoother and sharper (Figure 1c). However, compared with pure $Ag_3PO_4$, the particle length-to-width ratio is larger, and the rods ap-

pear to be more elongated. This result clearly indicates that BaO conjugation with $Ag_3PO_4$ changes the morphology of the pure $Ag_3PO_4$. Energy dispersive X-ray spectroscopy (EDX) was used to perform a quantitative element analysis of the samples. The EDX diagram of $Ag_3PO_4$ indicates that the main elements in the material are Ag, P, and O, and that no impurity elements are present, as shown in Figure 1b. This proves the formation of $Ag_3PO_4$. The EDX diagram of BaO@$Ag_3PO_4$ reveals that the main elements are Ag, P, O, and Ba, and that no impurity elements are present, as shown in Figure 1d. This indicates the successful incorporation between BaO and $Ag_3PO_4$. A TEM image shows that the particle size of the BaO@$Ag_3PO_4$ composite is around 40 nm (Figure 1e), which matches the XRD result.

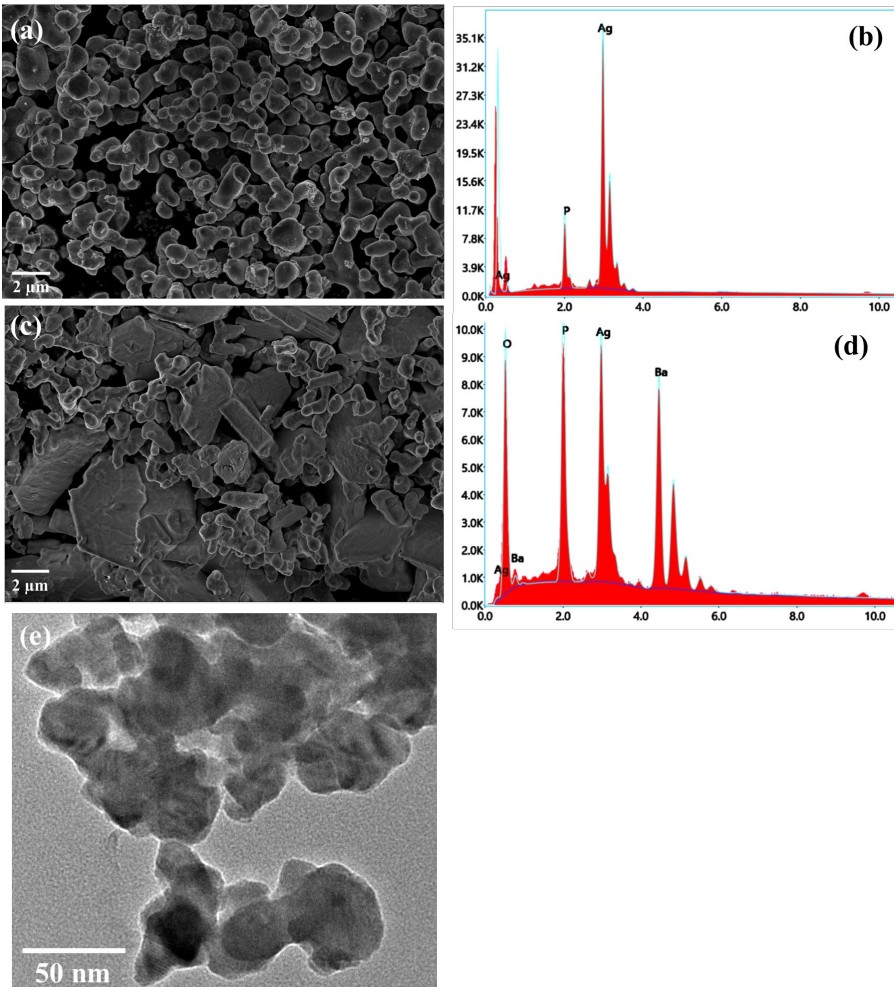

**Figure 1.** (**a**) Top-view SEM image of $Ag_3PO_4$ sample and (**b**) its EDX spectroscopy; (**c**) SEM image of BaO@$Ag_3PO_4$ composite and (**d**) its EDX spectroscopy; and (**e**) TEM image of BaO@$Ag_3PO_4$ composite.

2.1.2. X-ray Diffraction (XRD) Analysis

The XRD spectrum of the pure $Ag_3PO_4$ and the BaO@$Ag_3PO_4$ nanocomposite in the present study is shown in Figure 2. In the XRD pattern of the pure AP, the main diffraction peaks are shown at the 2θ positions [20.37°, 29.32°, 32.90°, 36.25°, 47.89°, 52.37° and 54.60°], which corresponds to plans (110), (200), (210), (211), (220), (310), and (222), those indexed to the pure body-centered cubic structure of AP according to the standard spectrum JCDPs No. (01-074-1876). In addition, the XRD patterns of the APB nanocomposite show peaks at the 2θ positions [24.91°, 27.19°, 29.61°, 31.90°, 33.57°, 36.57°,42.40°, 46.76°, 52.71°and 55.01°] of the synthesized material. The peaks were indexed by comparing them with the standard data available, JCDPs No. (00-033-16167), which confirmed the successful synthesis of the BaO@$Ag_3PO_4$ composite, and no impurity diffraction peaks were observed. The average

crystallite sizes of the samples have been calculated from the full width at half maximum (FWHM) of the XRD pattern in Figure 2, using the Williamson–Hull formula [21],

$$\beta \times \cos(\theta) = [K \times \lambda/D] + [4 \times S \times \sin(\theta)] \tag{1}$$

where $\beta$ is the FWHM in radians, $\lambda$ is the wavelength of the X-rays used, $\theta$ is the scattering Bragg angle, D is the particle diameter, and S is the strain. The parameter K is the shape factor (taken as 0.94). The crystallite sizes of the samples were 39.1 and 40 nm for the pure AP sample and the APB composite, respectively.

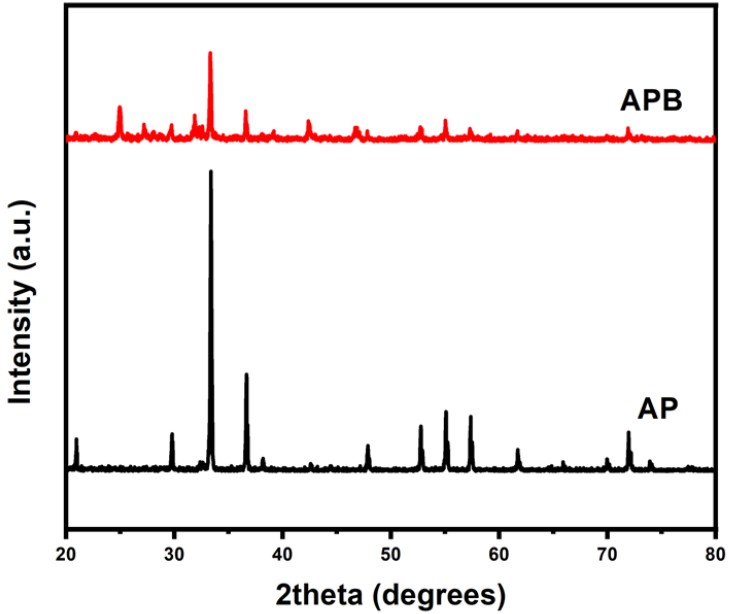

**Figure 2.** The powder X-ray diffraction patterns of pure $Ag_3PO_4$ and $BaO@Ag_3PO_4$ nanocomposite.

2.1.3. FT-IR Spectral Analysis

The FTIR spectra of the as-prepared sample, shown in Figure 3, were measured in the range of 400–4000 cm$^{-1}$. The spectra of the pure $Ag_3PO_4$ exhibit strong bands at 3178 and 1669 cm$^{-1}$ that may be attributed to the bending modes of the O-H groups (P–O–H bridges) and molecular water due to moisture absorption during the KBr pellet preparation [21]. The absorption band at 560 cm$^{-1}$ corresponds to the symmetric stretching modes of the P–O–P linkages, while the band at 901.87 cm$^{-1}$ represents the asymmetric stretching mode of P–O–P links [21]. The band appearing at 1010 cm$^{-1}$ is attributed to P–O− groups, which are vibrational and asymmetric stretching modes of the chain terminating $(PO_3)^{2-}$ [21]. In the presence of BaO, different vibrational bands are found to shift and present new bands at (534, 952, 734, 608, 489, and 431). This indicates that BaO changes the structure of the pure silver phosphate. On the basis of the FT-IR spectra, it can be inferred that, after the addition of BaO, the basic structure of $Ag_3PO_4$ is almost the same, but some P-O-Ag bonds present in the pure $Ag_3PO_4$ may be replaced by P-O-BaO bonds. It may be proposed that Ba2+ ions are attached to the negative end of the P-O bond. Thus, the conjugation metal ions form a P-O-Ba bond in the glass structure, and $Ba^{2+}$ ions serve as ionic cross-links between the non-bridging oxygen of two different phosphate chains [21].

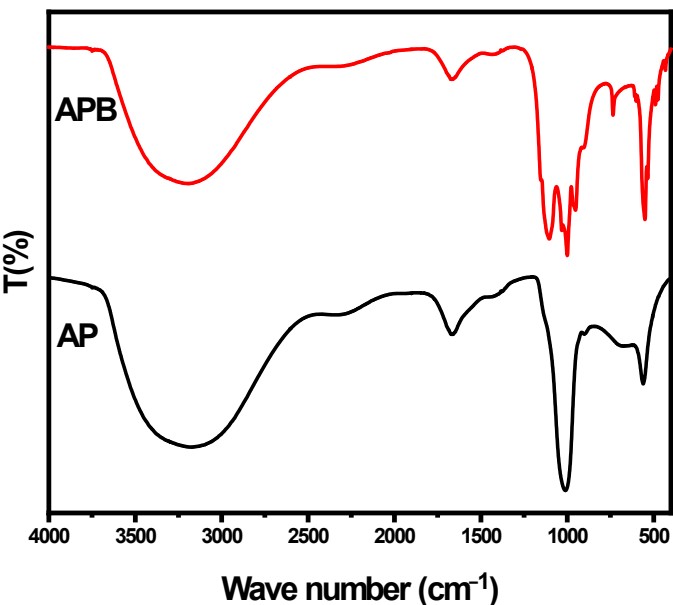

**Figure 3.** FT-IR spectra of pure $Ag_3PO_4$ and $BaO@Ag_3PO_4$ nanocomposite.

### 2.2. Optical Analysis

The optical properties of the prepared samples were investigated using UV–Vis DRS and PL spectra at room temperature.

#### 2.2.1. The DRS Spectra and Band Gap

There are several mechanisms that control photocatalytic activity: the production of electron–hole pairs, photo absorption, charge carrier transfer, and charge carrier utilization. The optimization of the photocatalytic activity depends on the efficiency of the product and the transfer of the e−/h+ pairs, which depends on the energy band gap (Eg) of the photocatalyst. The energy band gap values (Eg) of the samples were calculated according to the following Equation (2) from the reflectance curves [22,23]:

$$\alpha \, hv = [A(hv\text{-}Eg)]\hat{}(n/2) \tag{2}$$

where $\alpha$ is the absorption coefficient, v is the frequency of light, and n is the constant of proportionality. The n value is determined by the transition of the semiconductor, i.e., the direct transition as in the prepared nanocomposite (n = 1). The diffuse reflection spectra of the as-prepared $Ag_3PO_4$ and $BaO@Ag_3PO_4$ composite were investigated using UV–Vis optical spectroscopy in the range of 200–800 nm, as shown in Figure 4. It is observed that the pure $Ag_3PO_4$ could absorb visible light as shown in Figure 4a, in agreement with the results previously reported [21]. In contrast, the $BaO@Ag_3PO_4$ composite has a higher absorbance than $Ag_3PO_4$ in the range of 500–800 nm. The band gap value of pure $Ag_3PO_4$ is 2.43 eV. However, the band gap of the $BaO@Ag_3PO_4$ nanocomposite is 2.36 eV shifted towards the red shift in the presence of barium oxide. The optical band gaps were calculated from (F(R) hv)1/2 against the photon energy (hv) plots as showed in Figure 4b. The conjugation of the two band gaps led to more stability between the e−/h+ pairs, as the band gap reduces the quantum confinement effect. Moreover, $BaO@Ag_3PO_4$ nanocomposites can be excited under visible light to give more electron–hole pairs. This remarkable absorption enhancement in the visible light region is beneficial for improving the photocatalytic activity in this irradiation region.

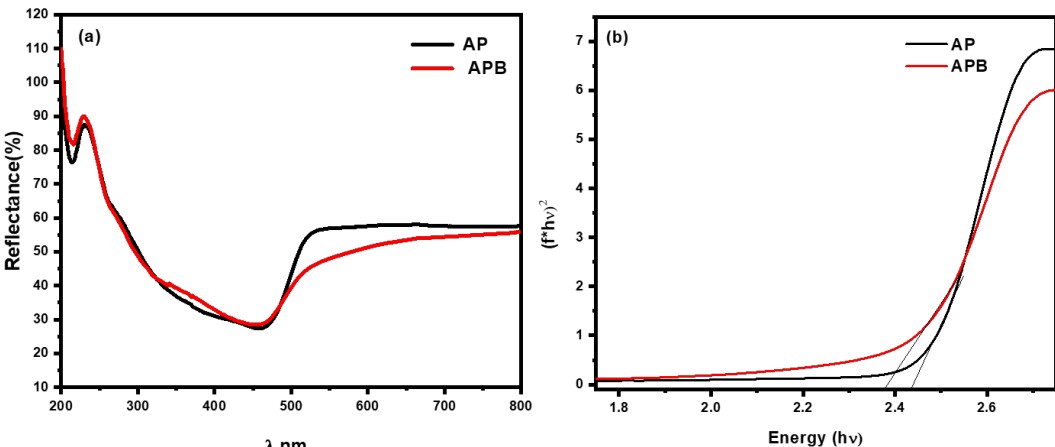

**Figure 4.** (**a**) UV–Vis diffuse reflectance spectra of pure $Ag_3PO_4$ and $BaO@Ag_3PO_4$ nanocomposite. (**b**) Band gap spectra of pure $Ag_3PO_4$ and $BaO@Ag_3PO_4$ nanocomposite.

### 2.2.2. Photoluminescence (PL)

Figure 5 displays the room temperature photoluminescence (PL) spectra of the AP and the APB composite at an excitation wavelength of 370 nm. The spectra show the rate of recombination and charge separation within the photocatalysts. The PL spectrum of pure silver phosphate shows a strong emission band at 422 nm, while the APB composite shows a significant intensity decrease in PL as a result of the efficient charge transfer at the heterostructure interface, indicating that the recombination between the photo-electrons generated and the holes is reduced. Moreover, there is a synergistic effect between the AP and the BaO as a heterostructure that decreases the PL intensity, which leads to an increase in the lifetime of electron stability [21].

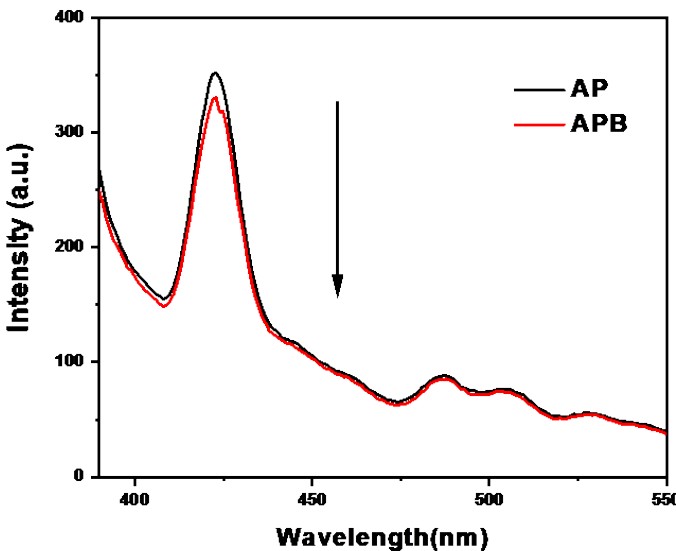

**Figure 5.** Photoluminescence spectra of pure AP and APB composite.

### 2.3. Photocatalytic Activity of APB Composite

The photocatalytic degradation of MB dyes under visible light caused by the pure $Ag_3PO_4$ and the $BaO@Ag_3PO_4$ composite is shown in Figure 6a. As presented in this figure, it can clearly be seen that the APB composite has a higher photocatalytic activity for the degradation of MB than the $Ag_3PO_4$ photocatalyst. Not only did the BaO act as an electron promoter and improve the charge separation of electrons and holes that was created by light irradiation, but the BaO also improved the stability of the electron–hole pairs. Figure 6a measures the degradation at times of irradiation of 0, 30, 60, 90, and 120 min. Before the

photocatalytic reaction, the photocatalyst's MB solution was kept for 30 min in the dark to reach the adsorption/desorption equilibrium. This equation gives the efficiency of the degradation of MB:

$$D\% = (C_0 - C/C_0) \times 100 \tag{3}$$

where $C_0$ is the initial concentration and $C$ is the remaining concentration of MB after the reaction. As shown in Figure 6a, the efficiency of the MB removal in the presence of pure AP and of APB is about 88% and 94%, respectively, after 120 min. According to the L–H kinetics model, the kinetics of the MB degradation by the prepared nanocatalysts was evaluated. The equation of pseudo-first-order kinetics can be expressed as

$$\ln(C_0/C) = k_a \times t \tag{4}$$

where $k_a$ is the rate constant ($min^{-1}$), $C_0$ is the initial concentration ($mg\ L^{-1}$), and $C$ is the concentration of the MB solution at the irradiation time $t$ (min). Figure 6b shows the relation between $\ln(C_0/C)$ and time. $K_a$ can be obtained from the linear relation between them, and it is ($3.137 \times 10^{-4}$, $0.02295$ and $0.02996\ min^{-1}$) for MB, AP, and APB, respectively. The rate constants are increased in the following order: APB >AP >MB. Based on these results, the APB composite shows the best activity. These characteristics increase their photocatalytic activity under visible light, and this is proved by their enhanced degradation of MB under visible light irradiation. Therefore, the as-prepared APB composites can work as effective photocatalysts for organic compound degradation with good stability. In addition, as shown in Table 1, $BaO/Ag_3PO_4$ had the highest photocatalytic activity under visible light illumination in comparison with the results of earlier studies.

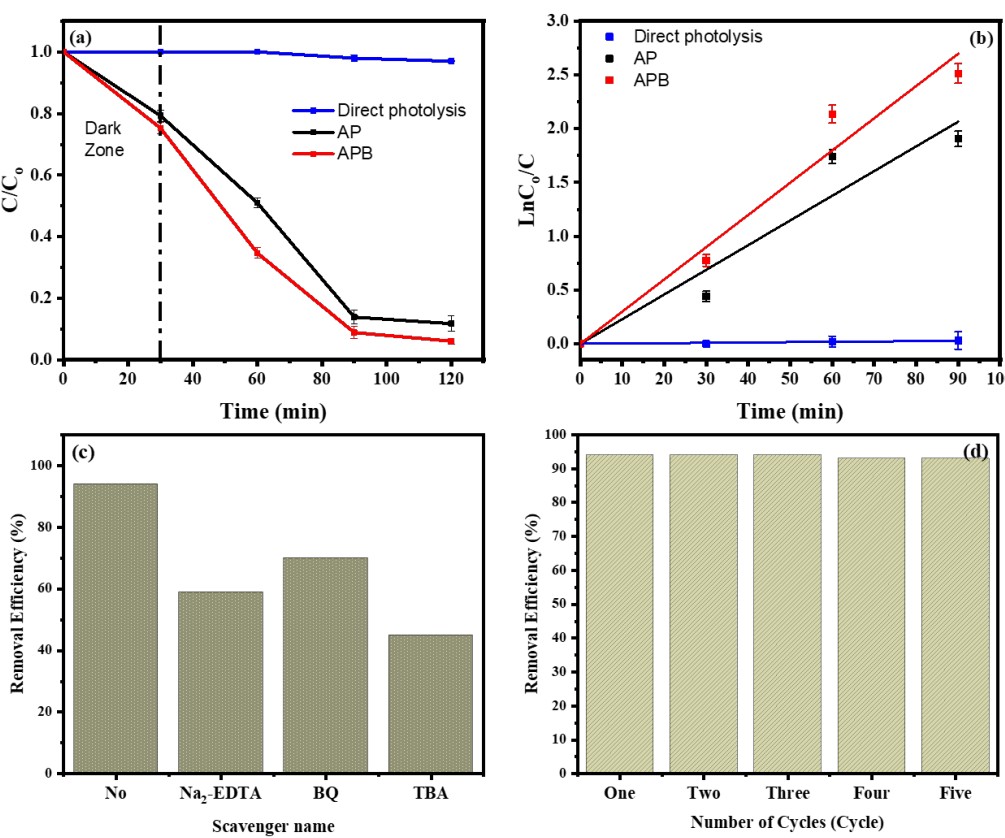

**Figure 6.** (**a**) The removal curve of MB in the presence of pure $Ag_3PO_4$ and $BaO@Ag_3PO_4$ nanocomposite. (**b**) Kinetics of MB removal by pure $Ag_3PO_4$ and $BaO@Ag_3PO_4$ nanocomposite. (**c**) The influence of different scavengers on the removal efficiency of MB by $BaO@Ag_3PO_4$ and (**d**) the removal efficiency of $BaO@Ag_3PO_4$ as a catalyst for MB degradation throughout five consecutive cycles of filtration and reuse (after 2 h).

In addition, we used several free radical trapping agents with the BaO@Ag₃PO₄ nanocomposite as described to make inferences about the roles of hydroxyl radicals (•OH), holes (h⁺), and superoxide anions (•O₂⁻) in the efficiency of the photodegradation of MB (Figure 6c). The trapping agents were Tert-butyl alcohol (TBA), disodium ethylene-diaminetetraacetic acid (Na₂-EDTA), and p-benzoquinone (BQ), which were used to trap free radicals of hydroxyl radicals (•OH), holes (h⁺), and superoxide radicals (•O₂⁻), respectively [24,25]. The photodegradation of MB differed depending on the sacrificial agent. The photocatalytic degradation was reduced to 45% in the presence of TBA (5 mM). As a result, during the photodegradation of MB, the •OH radical that was created by the photo-process was crucial [26]. Additionally, as shown in Figure 6d, the BaO@Ag₃PO₄ nanocomposite exhibits strong stability in its MB removal for up to five cycles. After five cycles, the photodegradation of MB was still stable at 93%. These BaO@Ag₃PO₄ nanocomposite results show a strong photodegradation of MB with a narrow energy band gap as well as a remarkable stability of catalytic efficiency.

**Table 1.** Removal of MB under visible light, with various photo catalysts.

| Photocatalyst | Weight of Catalyst (g/L) | Concentration of MB (ppm) | Time (h) | Degradation (%) | Ref. |
|---|---|---|---|---|---|
| Mesoporous Pt/WO₃ | 0.5 | 3 | 1.2 | 63 | [27] |
| Fe₃O₄/CdWO₄/PrVO₄ | 0.02 | 25 | 2 | 68 | [28] |
| WO₃-GO | 0.5 | 3 | 1.2 | 82 | [27] |
| c-NaNbO₃ | 1.04 | 10 | 3 | 87 | [29] |
| Fe₃O₄/CdWO | 0.02 | 25 | 2 | 32 | [30] |
| BaO/Ag₃PO₄ | 0.2 | 20 | 1.5 | 94 | This study |

*2.4. Photocatalysis Mechanism for Degradation of Organic Pollutant by APB Composite*

The mechanisms of APB photocatalysts presented in this part are based on all of the characterizing results described previously. The presence of BaO enhanced the light absorption of the catalyst to be active in the visible region. It has a small band gap energy which is active in the visible region. In addition, the junction between Ag₃PO₄ and BaO enhanced the stability between the electron–hole pairs. Moreover, BaO plays an important role in capturing the electrons from the conduction bands of the Ag₃PO₄ to increase the lifetime of the electrons and, subsequently, enhance the photocatalytic activity, as recorded in this study. Thus, the mechanism of the APB photocatalysts after irradiation under visible light is summarized by the following equations (Equations (5) to (12)) and illustrated in schematic the graph in Figure 7.

The photocatalytic APB can be excited under visible light irradiation to generate electron–hole pairs (Equation (5)). The electrons in the CB of the Ag₃PO₄ were transferred to the surface of the BaO, where BaO acted as an electron acceptor and increased the formation of oxide radicals (•O₂⁻) through a reduction process at the conduction band (Equation (6)). These oxide radicals played important roles in the degradation of organic pollutants and the formation of hydroxyl radicals (Equation (7)). The holes in the valance band at Ag₃PO₄ formed hydroxyl radicals with water molecules according to Equations (8)–(11). Finally, the organic pollutant was degraded by the hydroxyl radicals into environmentally friendly molecules: CO₂ and H₂O (Equation (12)).

$$Ag_3PO_4 + BaO + h\nu \quad \rightarrow \quad h^+ + e^- \tag{5}$$

$$Ag_3PO_4\,(e^-)/BaO + O_2 \quad \rightarrow \quad \bullet O_2^- \tag{6}$$

$$\bullet O_2^- + 2H_2O \quad \rightarrow \quad 4OH\bullet \tag{7}$$

$$h^+ + H_2O \quad \rightarrow \quad OH\bullet + H^+ \tag{8}$$

$$H_2O + H^+ + \bullet O_2^- \quad \rightarrow \quad H_2O_2 + OH\bullet \tag{9}$$

$$H_2O_2 + e^- \quad \rightarrow \quad OH^- + OH\bullet \tag{10}$$

$$h^+ + OH^- \quad \rightarrow \quad OH\bullet \tag{11}$$

$$OH\bullet + \text{organic pollutant} \rightarrow \text{degradation to } CO_2 + H_2O \tag{12}$$

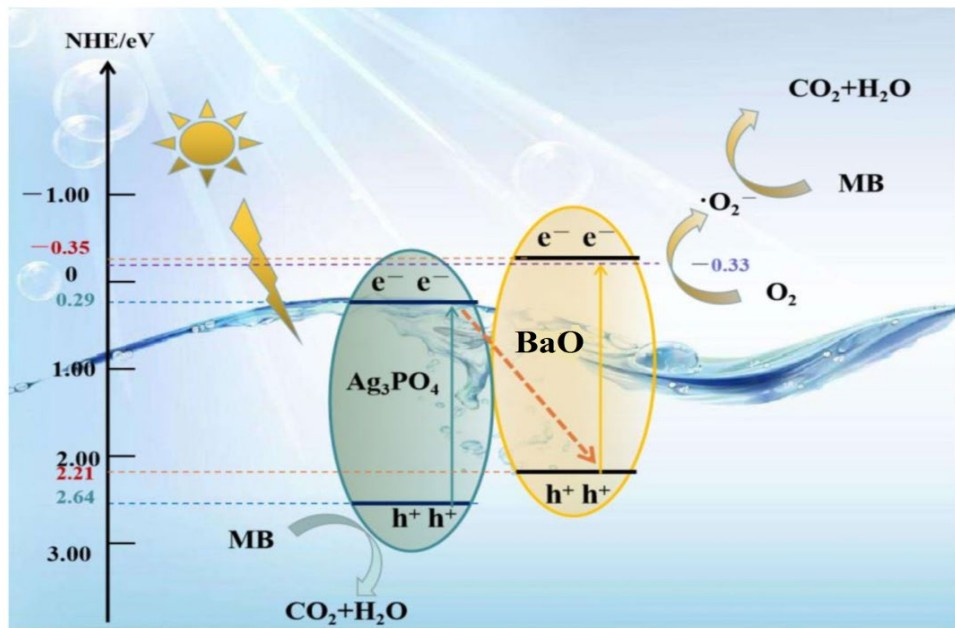

**Figure 7.** Schematic diagram of the photocatalytic mechanism of $BaO@Ag_3PO_4$ nanocomposite under visible light irradiation.

### 2.5. Hydrogen Evolution Detected by Prepared Photocatalysts

The photocatalytic $H_2$ evolution of the pure $Ag_3PO_4$ and the $BaO@Ag_3PO_4$ nanocomposite under visible light irradiation is shown in Figure 8. Figure 8 shows that the $BaO@Ag_3PO_4$ photocatalyst achieves an $H_2$-production rate of up to 490 μmol during 6 h under visible light irradiation. It is found that the photocatalytic activity of the composites is highly related to the content of the BaO and the synergism effect between the BaO and the $Ag_3PO_4$. To describe the true hydrogen production efficiency of a water-splitting reaction under visible light, we detected the quantum efficiency of the photocatalysts, and it is 5.3 and 4.5% for $Ag_3PO_4$ and $BaO@Ag_3PO_4$, respectively. The quantum efficiency was calculated from the conversion efficiency according to the following equation [23]:

$$STH = \left[ \frac{(\text{mmole} H_2 / S) \times (237 KJ/mol)}{P_{total}(mW/cm^2) \times Area(cm^2)} \right] \tag{13}$$

where $H_2$ is the hydrogen evolution rate, which was multiplied by the Gibbs free energy for generating one mole of hydrogen from water; $P_{total}$ is the total power of the irradiation light; and Area is the area which was exposed to the incident light. Compared to earlier investigations of other nanocomposite photocatalysts, the obtained nanocomposite $BaO@Ag_3PO_4$ produces hydrogen with strong activity, as shown in Table 2.

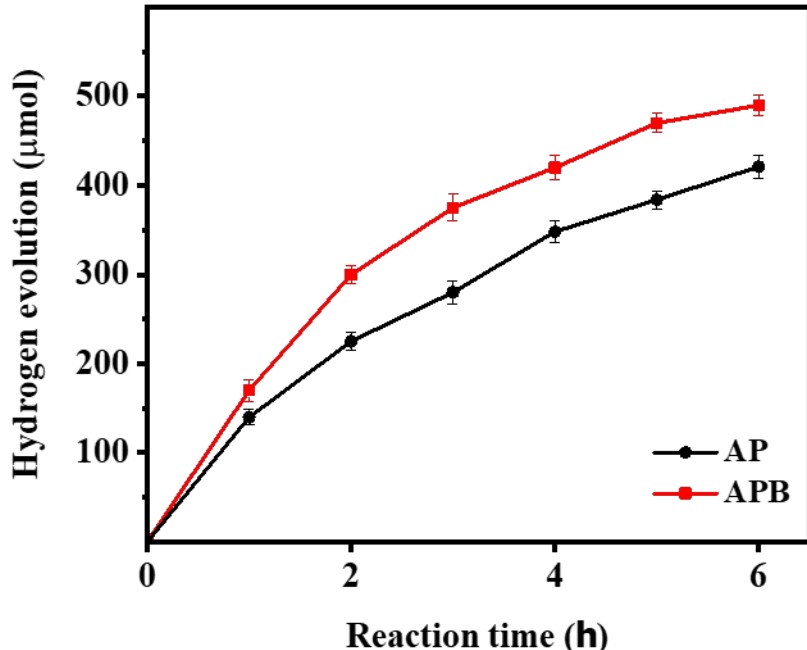

**Figure 8.** Comparison of the photocatalytic activity of the pure $Ag_3PO_4$ and $BaO@Ag_3PO_4$ nanocomposite samples for photocatalytic hydrogen production.

**Table 2.** Photocatalytic activity of various photocatalysts.

| Photocatalyst | $H_2$ Evolution Rate ($\mu mol/(h.g)$) | Ref. |
|---|---|---|
| $Ag_3PO_4$ -glass nanocomposite | 3921 | [31] |
| CdS nanowires | 3700 | [32] |
| CdS-$Ag_3PO_4$ hetero-nanostructures | 6600 | [32] |
| $Ag_3PO_4$–$TiO_2$ | 453 | [33] |
| BN/$Ce_2O_3$/$TiO_2$ nanofibers | 850 | [34] |
| BaO/$Ag_3PO_4$ | 7538 | This study |

## 3. Materials and Methods

### 3.1. Chemicals

All of the chemicals used in this study were of analytical grade and were purchased from Sigma Aldrich (Cairo, Egypt). Silver nitrate ($AgNO_3$), Ammonium dihydrogen phosphate ($NH_4H_2PO_4$), Barium carbonate ($BaCO_3$), and NaOH powder were obtained from Sigma-Aldrich (Cairo, Egypt). Methylene blue (MB) dye was purchased from Merck KGaA (Frankfurter, Germany). All solutions were prepared using freshly deionized water. All chemicals were of analytical grade and were used as received from the supplier.

### 3.2. Methods

3.2.1. Synthesis of $Ag_3PO_4$ Nanoparticles

The $Ag_3PO_4$ nanoparticles (NPs) sample was synthesized using the co-precipitation method according to the following steps: An appropriate amount of $AgNO_3$ salt (0.5 gm) was dissolved completely in 50 mL of deionized water (DI) under continuous stirring at room temperature. An equal volume of $NH_4H_2PO_4$ crystalline powder (0.5 gm) was dissolved in 50 mL of deionized water (DI) under continuous magnetic stirring at room temperature [18]. Then, the first solution containing the $AgNO_3$ salt was added slowly to the second solution containing $NH_4H_2PO_4$ under continuous stirring. After that, the resulting solution was sonicated for 15 min to avoid any agglomeration. As a result, a yellow precipitate was obtained and then collected and washed several times with distilled

water. Finally, the precipitate was dried in an oven at 100 °C overnight, and then calcined at 400 °C for 1 hr to obtain the $Ag_3PO_4$ nanocrystalline sample [23].

### 3.2.2. Preparation of BaO@$Ag_3PO_4$ Composite

The BaO@$Ag_3PO_4$ composite was synthesized at room temperature using a co-precipitation technique through the following steps: An appropriate amount of $BaCO_3$ salt (0.5 gm) was dissolved completely in 50 mL of deionized water (DI) under stirring at room temperature. Next, 0.25 gm of $AgNO_3$ salt was added to 0.25 gm of $NH_4H_2PO_4$ and dissolved in 50 mL of deionized water (DI) under stirring at room temperature. The first solution containing $BaCO_3$ salt was added dropwise, in a slow manner, to the second solution under vigorous stirring at room temperature. Later on, the solution was sonicated for 15 min to avoid any agglomeration. After that, the obtained yellow precipitate was collected and washed several times with distilled water, then dried in an oven at 100 °C overnight, and then ground and calcined at 400 °C for 1 h to obtain the BaO@$Ag_3PO_4$ composite [18].

### 3.2.3. Characterization Techniques

The crystallographic patterns of the prepared nanocomposite were investigated using Philips X'Pert X-ray diffraction (XRD) with Cu-Ka radiation (λ = 1.54056° A) and an accelerating voltage of 40 kV and 40 mA. A typical 2θ scan was between 10–80° and conducted at a scan rate of 4°/min (Panalytical, Almelo, Netherlands). Fourier-transform infrared spectroscopy (FTIR) was performed using a Nicolet iS10 FTIR spectrometer to identify the surface functional groups in the synthesized materials (Perkin Elmer, Waltham, MA, USA). The morphological features were evaluated by scanning electron microscopy (SEM, Hitachi S-4800, Boise, USA). Ultraviolet-visible spectroscopy (UV–Vis, Jasco V-507) coupled with a Shimadzu IRS-2200 (Perkin Elmer Lambda 1050,Waltham, MA, USA) diffuse reflectance spectroscopy (DRS) was used to investigate the optical properties of the prepared materials. Solid-state photoluminescence (PL) analysis was also performed using a Perkin Elmer fluorometer (model LS-55) at room temperature.

### 3.2.4. Photocatalytic Activity Study and $H_2$ Production

The photocatalytic activity of the synthesized nanocomposite was evaluated for the degradation of the MB dye as a model of organic pollutants under visible light irradiation. The experiments were carried out under visible light irradiation at room temperature by a halogen lamp with a power of 500 W, and the distance from the light source to the cell was 10 cm, with 80 mW/cm$^2$ measured by X1-1 Optometer (Gigahertz-Optik) RW-3705-4 (400–1000 nm) calibrated detectors. In each experiment, 50 mL of MB dye solution with an initial concentration of 20 ppm was taken in a glass beaker, and 2.0 g/L of the prepared catalyst was added.

The solution was stirred in the darkness for 30 min until it reached an adsorption/desorption equilibrium between the MB dye molecule and the catalyst. Then, the experiment was carried out further with constant stirring under visible light irradiation. After that, we collected 3 mL of the solution 4 times every 30 min, and we separated the catalyst by centrifuge and then detected the concentration using a UV–visible spectrophotometer (UV-1800, Shimadzu, Kyoto, Japan) at the maximum absorption wavelength (622 nm). In order to test the catalyst's activity, a dark reaction was performed. In the absence of the catalyst, a blank experiment was carried out, and it was observed that the dye degradation was limited.

For the $H_2$ evolution test, 10 mg of photocatalyst was dispersed with a constant stirring in 50 mL of an aqueous solution of sacrificial reagent (0.1 M $Na_2S$ + 0.02 M $Na_2SO_3$). After a given irradiation time, the evolved hydrogen was collected and monitored using gas chromatography (micro-GCAgilent, Santa Clara, California, United States) with a 5 Å molecular sieve column (3 m × 2 mm). Each experimental run was performed in triplicate to observe the reproducibility, so the photocatalytic results presented in this work are the average of three replicates.

## 4. Conclusions

In summary, the APB composite has been successfully synthesized using a simple co-precipitation method as a novel catalyst for methylene blue (MB) degradation. Afterwards, the physicochemical properties of the resulting samples were studied through a series of characterizations. Through these characterizations, it was found that the addition of BaO slightly changed the morphology of the material; additionally, it greatly improved its bandgap structure and improved the range of its response to visible light. From UV–visible studies, it was shown that the optical band gaps are 2.43 and 2.36 eV for the $Ag_3PO_4$ and the APB, respectively, which indicates that the APB absorbs in both the UV and visible regions. Additionally, the composite has a high separation efficiency for the photo-generated electron and hole, and the major active species engaged in the reaction are $O_2^-$ and h+. Furthermore, the as-synthesized photocatalyst exhibited excellent photocatalytic efficiency. Under visible light irradiation, nearly 94% of the MB (20 mg $L^{-1}$, 50 mL) can be degraded by only 0.02 g of the catalyst in 120 min. In addition, the enhanced visible light mechanism of the $Ag_3PO_4$/BaO composite was proposed. Overall, a higher photocatalytic activity for MB degradation is obtained for the APB photocatalyst than for the $Ag_3PO_4$ photocatalyst, which is ascribed to the highly synergistic charge transfer between the BaO and the $Ag_3PO_4$ support. On the basis of these results, it can be concluded that the BaO@$Ag_3PO_4$ composite is a promising photocatalyst candidate for the photodegradation of hazardous organic materials in wastewater.

**Author Contributions:** Conceptualization, H.S., E.R.S. and A.A.N.; methodology H.S., R.E. and H.H.E.-M.; validation, all authors; formal analysis, H.S., R.E. and H.H.E.-M.; investigation, H.S, E.R.S., H.H.E.-M., P.R. and A.A.N.; resources, E.R.S. and H.S.; data curation, H.S. and A.A.N.; writing—review and editing, H.S., E.R.S., H.H.E.-M., R.E., P.R. and A.A.N.; visualization H.S., E.R.S., R.E., P.R., H.H.E.-M. and A.A.N.; supervision, A.A.N. and H.S. All authors have read and agreed to the published version of the manuscript.

**Funding:** This research received no external funding.

**Data Availability Statement:** Not applicable.

**Conflicts of Interest:** The authors declare no conflict of interest. The funders had no role in the design of the study; in the collection, analyses, or interpretation of data; in the writing of the manuscript; or in the decision to publish the results.

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
