# Peer review of "Superior Photocatalytic Activity of BaO@Ag3PO4 Nanocomposite for Dual Function Degradation of Methylene Blue and Hydrogen Production under Visible Light Irradiation"

_catalysts, doi:10.3390/catal13020363_

Round 1
Reviewer 1 Report
Manuscript #: catalysts-2096949
Title: Superior Photocatalytic Activity of BaO@Ag3PO4 Nanocomposite for Dual Function Degradation of Methylene Blue and hydrogen Production under Visible light irradiation.
Authors: Selim, Sheha, Elshypany, EI-Maghrabi, and Nada.
In this manuscript, barium oxide/sliver phosphate BaO@Ag3PO4 as a photocatalyst was synthesized by chemical co-precipitation. The synthesized photocatalysts were used as degradation of methylene blue (MB) and hydrogen production. The characteristics of produced catalyst were analyzed with SEM, XRD, UV-DRS, and FT-IR.
The topic of this manuscript is interesting, and this study shows interesting results to apply this technology for decomposition of MB and hydrogen production. However, several contents in the manuscript require more clarity for publication.
1) Incorporation of element such as π – conjugation by BaO must be proven with XPS analysis.
2) The intensity of visible light must be measured and added with its unit of mW/cm2 or W/m2.
3) Although the size of crystalline was calculated using the results of XRD analysis, it would be better to confirm it with TEM images of crystal size.
4) The adsorption equilibrium data must be presented. Based on the manuscript, the degradation of MB looks difficult to know if it is removed by adsorption or photocatalytic effect. Due to photosensitization effects by dyes for visible light induced photocatalysis, different contaminants, which do not show any photosensitization, must be used to verify the synthesis of visible light active photocatalysts.
5) Quenching test (electron hole separation) should be conducted due to investigate the reaction mechanism of the photocatalysis.
6) The reusability of the catalysts must be tested. Also, Ba and Ag leaching from the catalysts must be checked during the experiments.
7) In Table 1, it should be good to show other conditions such as initial pH, concentration, and load of catalysts.
Therefore, I recommend “Major revision” of this manuscript for publication.
Author Response
Reviewer 1:
- Incorporation of element such as π –conjugation by BaO must be proven with XPS analysis.
Response 1: Thank you for decision and We highly appreciated the time you devoted to a careful revised the manuscript. We detect the structure and the morphology of the prepared materials by the available techniques (FTIR, XRD, UVDR, SEM) and we enhanced the revised manuscript by more analysis as TEM as recommended by reviewer.
- The intensity of visible light must be measured and added with its unit of mW/cm2or W/m2.
Response 2: we added the intensity of visible light in the manscript. “With 80 mW/cm2 was measured by X1-1 Optometer (Gigahertz-Optik) RW-3705-4 (400–1000 nm) calibrated detectors.”
- Although the size of crystalline was calculated using the results of XRD analysis, it would be better to confirm it with TEM images of crystal size.
Response 3: we modified the manuscript by detect the TEM image as recommended by reviewer.
- The adsorption equilibrium data must be presented. Based on the manuscript, the degradation of MB looks difficult to know if it is removed by adsorption or photocatalytic effect. Due to photosensitization effects by dyes for visible light induced photocatalysis, different contaminants, which do not show any photosensitization, must be used to verify the synthesis of visible light active photocatalysts.
Response 4: We detected the photolysis of MB without the catalyst under the same light source is less than 3% as presented in Figure 6 (direct photolysis). Moreover, we detected the hydrogen yield by the photocatalytic effect, which is more important and is a good sign of the efficiency of the prepared catalyst.
- Quenching test (electron hole separation) should be conducted due to investigate the reaction mechanism of the photocatalysis.
Response 5: We added different scavengers to investigate the reaction mechanism of the photocatalysis in revised manuscript (page 12) as recommended.
- The reusability of the catalysts must be tested. Also, Ba and Ag leaching from the catalysts must be checked during the experiments.
Response 6: We added the reusability of the catalysts in revised manuscript (page 12) as recommended.
- In Table 1, it should be good to show other conditions such as initial pH, concentration, and load of catalysts.
Response 7: We added more conditions test of our study in revised manuscript (page 12) as recommended. but for the Table 1, we focused on the striking results available in the literature studies to compare with our study.

Reviewer 2 Report
Section Results and discussion can be improved. For example, it is not enough to speculate within the degradation mechanism. It is need to investigate the influence of different scavengers on the degradation efficiency of the dye. In addition, it is necessary to examine the reusability of newly synthesized photocatalyst through at least three cycles.
In Conclusion states that "much higher photocatalytic activity for MB degradation is obtained in the APB photocatalyst than the Ag3PO4 photocatalyst". However, difference between pure AP and APB efficiency is only about 6%. Besides, is it 60 or 120 min of removal (row 273)? It is necessary to check the time of process in different sections of manuscript. Furhermore, it is need to pay attention that it is NOT only the process of irradiation/degradation (for example 0, 30, 60 min, etc.), and accordingly instead of degradation it should be used terms process or removal. Please check carefully through manuscript.
Figure caption "Figure 6. (a) The photocatalytic degradation curve of pure Ag3PO4 and BaO@Ag3PO4 nanocompositep (b) Kinetic of MB degradation by pure Ag3PO4 and BaO@Ag3PO4 nanocomposite" is not correct because the MB is removed, not photocatalysts. It should be state "Figure 6. (a) The removal curve of MB in the presence of pure Ag3PO4 and BaO@Ag3PO4 nanocomposite. (b) Kinetics of MB removal by pure Ag3PO4 and BaO@Ag3PO4 nanocomposite. Also, in row 272 part of sentence: "The efficiency of pure AP and APB degradation..." is wrong and it should be state: "The efficiency of MB removal in the presence of pure AP and APB..." Please check carefully through manuscript.
Legend in the Fig. 6 (black curve) is also better to mark as "direct photolysis".
Author Response
Reviewer 2
Comment1: Section Results and discussion can be improved. For example, it is not enough to speculate within the degradation mechanism. It is need to investigate the influence of different scavengers on the degradation efficiency of the dye. In addition, it is necessary to examine the reusability of newly synthesized photocatalyst through at least three cycles.
Response 1: Thank you for decision and We highly appreciated the time you devoted to a careful revised the manuscript. The influence of different scavengers on the degradation efficiency of the dye and the reusability of newly synthesized photocatalyst through five cycles added in revised manuscript (page 12).
Comment 2: In Conclusion states that "much higher photocatalytic activity for MB degradation is obtained in the APB photocatalyst than the Ag3PO4 photocatalyst". However, difference between pure AP and APB efficiency is only about 6%. Besides, is it 60 or 120 min of removal (row 273)? It is necessary to check the time of process in different sections of manuscript. Furhermore, it is need to pay attention that it is NOT only the process of irradiation/degradation (for example 0, 30, 60 min, etc.), and accordingly instead of degradation it should be used terms process or removal. Please check carefully through manuscript
Response 2:
- The conclusion about "much higher photocatalytic activity for MB degradation is obtained in the APB photocatalyst than the Ag3PO4 photocatalyst" modified to be" higher photocatalytic activity for MB degradation is obtained in the APB photocatalyst than the Ag3PO4 photocatalyst" in line 469.
- The time of removal modified to be 120 min. in line 273.
- The of time irradiation (0, 30, 60, 90, and 120) min stated in line 265.
- The efficiency of MB degradation in the presence of pure AP and APB modified to be "The efficiency of MB removal in the presence of pure AP and APB" in lines 272 and 273.
Comment 3: Figure caption "Figure 6. (a) The photocatalytic degradation curve of pure Ag3PO4 and BaO@Ag3PO4 nanocompositep (b) Kinetic of MB degradation by pure Ag3PO4 and BaO@Ag3PO4 nanocomposite" is not correct because the MB is removed, not photocatalysts. It should be state "Figure 6. (a) The removal curve of MB in the presence of pure Ag3PO4 and BaO@Ag3PO4 nanocomposite. (b) Kinetics of MB removal by pure Ag3PO4 and BaO@Ag3PO4 nanocomposite. Also, in row 272 part of sentence: "The efficiency of pure AP and APB degradation..." is wrong and it should be state: "The efficiency of MB removal in the presence of pure AP and APB..." Please check carefully through manuscript.
Response 3: We corrected as recommended by reviewer.
Comment 4: Legend in the Fig. 6 (black curve) is also better to mark as "direct photolysis".
Response 4: We corrected as recommended by reviewer.

Round 2
Reviewer 1 Report
Authors appropriately revised their manuscript according to my comments. Therefore, I recommend "Accept" for publication.
Author Response
On behalf of my co-authors, I would like to thank you sincerely for your the Acceptance decision regarding our manuscript . We highly appreciated the time you devoted to a careful reading of the manuscript.
Reviewer 2 Report
The manuscript has serious flaws and unfortunately my recommendation is rejection.
The discussion that was added (lines 344-358) was written completely amateurishly, without any explanations. Also, in Figure 9c (which the authors marked by mistake as 7(c) in the discussion), on the y-axis is an intensity that is completely wrong, which indicates that the experiments are not well thought out methodologically and the results were not presented correctly. Besides, in the case of Figure 9d, on x-axis is presented 2 Theta in degrees, which is also completely wrong. In addition, not all mistakes have been corrected, for example the text below Figure 6, etc.
Author Response
Response: Thank you for comments.
-We revised the manuscript carefully and we corrected the discussion (lines 344-358, old version) as the following: (lines 240-253 in new version).
“In addition, we used several free radical trapping agents with BaO@Ag3PO4 nanocomposite as described to make inferences on the roles of hydroxyl radicals (•OH), holes (h+), and superoxide anions (•O2-) in the efficiency of photodegradation of MB (Figure 6c). The trapping agents were Tert-butyl alcohol (TBA), disodium ethylenediaminetetraacetic acid (Na2-EDTA) p-benzoquinone (BQ) to trap free radicals of hydroxyl radicals (•OH), holes (h+) and superoxide radicals (•O2-), respectively [24] [25]. The photodegradation of MB differed depending on the sacrificial agent. The photocatalytic degradation was reduced to 45% in the presence of TBA (5 mM). As a result, during the photodegradation of MB, the •OH radical that was created by the photo-process was crucial [26]. Additionally, as shown in Figure 6d, the BaO@Ag3PO4 nanocomposite exhibits strong stability of MB removal for up to five cycles. After five cycles, the photodegradation of MB was still stable at 93%. These BaO@Ag3PO4 nanocomposite results show a strong photodegradation of MB with a narrow energy band gap as well as remarkable stability of catalytic efficiency.”
-We corrected the typography error in Figure 9c and 9d (previous version of the manuscript). As the following: (Figure 6 c and d in new version)
-We corrected the text below figure 6 (the figure caption) as recommended by reviewer, as the following:
Figure 6. (a) The removal curve of MB in the presence of pure Ag3PO4 and BaO@Ag3PO4 nanocomposite. (b) Kinetics of MB removal by pure Ag3PO4 and BaO@Ag3PO4 nanocomposite.
